# Antibody reliability influences observed mRNA–protein correlations in tumour samples

Swathi Ramachandra Upadhya[1,2,3] , Colm J Ryan[1,2,3]

Reverse phase protein arrays (RPPA) have been used to quantify the abundance of hundreds of proteins across thousands of tumour samples in the Cancer Genome Atlas. By number of samples, this is the largest tumour proteomic dataset available and it provides an opportunity to systematically assess the correlation between mRNA and protein abundances. However, the RPPA approach is highly dependent on antibody reliability and approximately one-quarter of the antibodies used in the the Cancer Genome Atlas are deemed to be somewhat less reliable. Here, we assess the impact of antibody reliability on observed mRNA-protein correlations. We find that, in general, proteins measured with less reliable antibodies have lower observed mRNA–protein correlations. This is not true of the same proteins when measured using mass spectrometry. Furthermore, in cell lines, we find that when the same protein is quantified by both mass spectrometry and RPPA, the overall correlation between the two measurements is lower for proteins measured with less reliable antibodies. Overall our results reinforce the need for caution in using RPPA measurements from less reliable antibodies.

## Introduction

Proteins are the primary actors in our cells and their activities are often deregulated in cancer. Quantifying proteomic abundances in patient tumours can aid the identification of pathways and processes that are deregulated in specific tumours, whereas across cohorts of patients, variation in protein abundances can be linked with specific phenotypic outcomes and responses to therapy. Unfortunately, our ability to quantify protein abundances in large cohorts has lagged significantly behind our ability to quantify mRNA abundances. Consequently, mRNA abundances are frequently used as proxies for protein abundances. However, there is only a moderate correlation between the two measurements (Vogel & Marcotte, 2012; Liu et al, 2016; Buccitelli & Selbach, 2020), with this

limited correlation attributed to a combination of post-transcriptional regulation and limitations in our ability to accurately and reproducibly quantify transcripts and proteins (Vogel & Marcotte, 2012; Liu et al, 2016; Franks et al, 2017; Buccitelli & Selbach, 2020).

The two main approaches that have been employed to quantify protein abundances in patient cohorts are mass spectrometry (MS), which has been recently used to quantify the abundance of thousands of proteins in up to a couple of 100 patients (Ellis et al, 2013), and reverse phase protein arrays (RPPA), which have been used to quantify the abundances of a couple of 100 proteins in thousands of patients (Li et al, 2013; Chen et al, 2019). In recent work, we have shown how limitations in our ability to reproducibly quantify proteins using MS contribute to the moderately observed correlation between mRNA and protein abundances (Upadhya & Ryan, 2022). Here, we focus our analysis on technical factors associated with protein abundances measured using RPPA.

RPPA is a simple and cost-effective antibody-based quantification approach that can be used to profile large numbers of samples. The Cancer Genome Atlas (TCGA) program has used RPPA to quantify the abundances of ~200 proteins for >7,500 tumour samples across 32 different cancer types (Chen et al, 2019). In addition to RPPA profiles, most of TCGA samples have linked exome sequencing, copy number profiles, and transcriptomes, facilitating integrated analyses of different molecular profiles. Among other applications, the resulting profiles have been used to classify tumours into cancer types (Zhang et al, 2015), to understand the differential activation of signalling pathways across tumours (Zhang et al, 2017), to identify mechanisms of epithelial–mesenchymal transition (Koplev et al, 2018), and to assess the correlation between mRNA and protein abundances (Chen et al, 2019). Although the number of proteins quantified in this resource is relatively small, the large number of samples with linked RPPA profiles and transcriptomes provides a unique resource for understanding how mRNA variation contributes to protein variation.

A key factor in the accuracy of RPPA profiles is the reliability of the antibodies used, as even minor non-specific binding of antibodies can distort the signal from the target protein (Mannsperger et al, 2010). Antibody quality is a challenge for many biological assays, including Western blotting, but the problem is more acute

[1]School of Computer Science, University College Dublin, Dublin, Ireland   [2]Conway Institute, University College Dublin, Dublin, Ireland   [3]Systems Biology Ireland, University College Dublin, Dublin, Ireland

Correspondence: colm.ryan@ucd.ie

for RPPA studies (Mannsperger et al, 2010). In Western blots, the non-specific binding of antibodies can be somewhat addressed by focusing on the results corresponding to the expected molecular weight of the protein assayed, something which is not possible with RPPA. Furthermore, with many assays, the incubation conditions can be optimised for individual antibodies, but this is not possible with the hundreds of antibodies used in RPPA.

Although collectively they are often referred to as "high-quality antibodies" (Akbani et al, 2014b; Şenbabaoğlu et al, 2016; Cancer Genome Atlas Research Network, 2017; Zhang et al, 2017), the quality of the antibodies used for TCGA RPPA studies vary. All antibodies used are assessed by the MD Anderson Cancer Center (Li et al, 2013; Akbani et al, 2014b; Chen et al, 2019). The two minimum criteria for validating antibody specificity used by MD Anderson are (i) a single or dominant band in a Western blot around the expected molecular weight of the target protein and (ii) a good Pearson correlation (>0.7) between abundances measured by RPPA and Western blotting across multiple samples (Li et al, 2013; Akbani et al, 2014a). Based on these criteria, antibodies are either discarded as unfit for use, categorised as "Valid," or categorised as "Use with Caution." The antibodies marked as "Valid" indicate that they bind to the intended target protein, whereas the ones marked as "Use with Caution" indicate they may bind to off-targets or multiple proteins along with the target protein. Although the performance of "Use with Caution" antibodies is poorer than those categorised as "Valid," they are still used for quantification, typically because they bind to a protein known to have an important role in cancer.

To understand the influence of the reliability of antibodies on mRNA–protein correlation, we analyse studies containing mRNA and protein expression profiles from The Cancer Genome Pan-Cancer Atlas (TCGA Pan-Can) (Cancer Genome Atlas Research Network et al, 2013b), and the Clinical Proteomic Tumour Analysis Consortium (CPTAC) (Ellis et al, 2013). We find that proteins that are quantified using less reliable antibodies have lower mRNA–protein correlation when using RPPA measurements, whereas no such trend can be observed for the same proteins quantified using MS. These proteins do not appear to be less reliably quantified by MS nor are they overall less abundant. By analysing data in cancer cell lines, we find that proteins measured using less reliable antibodies tend to have a lower correlation with MS measurements of the same proteins. Overall, our results are consistent with antibody reliability influencing the accuracy of protein abundance measurements and reinforce the need for caution when analysing RPPA measurements made with less reliable antibodies.

## Results

### Proteins quantified using RPPA with valid antibodies have a higher mRNA–protein correlation

Approximately one-quarter (27%) of the antibodies used for TCGA RPPA studies are labelled as "Use with Caution" (Li et al, 2013). The reliability of the antibody used to quantify protein abundances will impact all downstream analyses of protein measurements including the analysis of mRNA–protein correlations. To understand

the impact of antibody reliability on observed mRNA–protein correlations, we obtained RPPA measurements from TCGA Pan-Cancer study (Thorsson et al, 2018). This dataset contains measurements for 258 proteins and phosphoproteins. However, the antibody reliability information is available for only 187 proteins and phosphoproteins (Li et al, 2013). Among these 187 antibodies, we further restricted our analyses to those that are annotated as measuring the abundance of a single, non-phosphorylated protein (See the Materials and Methods section). Of the 114 antibodies in this category, 34 are labelled as "Use with Caution," whereas 80 are labelled as "Valid," that is, ~30% of antibodies should be used with caution (Fig S1 and Table S1). The "Use with Caution" antibodies in this set include antibodies that bind the protein products of frequently altered cancer driver genes, such as the oncogenes MYC and BRAF and the tumour suppressors BRCA2 and VHL.

To assess the mRNA–protein correlation with respect to antibody validation status, we analysed six TCGA PanCancer Atlas studies with the highest number of samples profiled by RPPA—breast cancer (Cancer Genome Atlas Network, 2012b), ovarian cancer (Cancer Genome Atlas Research Network, 2011), colorectal adenocarcinoma (Cancer Genome Atlas Network, 2012a), endometrial carcinoma (Cancer Genome Atlas Research Network et al, 2013a), kidney renal cell carcinoma (Cancer Genome Atlas Research Network, 2013), and low-grade glioma (Cancer Genome Atlas Research Network et al, 2015). We quantified the mRNA–protein correlation for all individual, non-phosphorylated proteins measured by RPPA and found that the median mRNA–protein correlation for proteins with "Valid" antibodies was significantly higher compared with proteins with antibodies marked as "Use with Caution" (P-value ≤ 0.01 in all cases, two-sided Mann–Whitney U test) (Fig 1). We note that here, we are referring to correlations calculated for an individual protein across all samples within a study ("across-sample" correlation) (Vogel & Marcotte, 2012; Liu et al, 2016; Franks et al, 2017; Buccitelli & Selbach, 2020).

To assess if mRNA–protein correlations for different antibodies were consistent across studies, we computed Pearson's correlation between the mRNA–protein correlations measured in each pair of TCGA studies. We found that the average correlation between all pairs of studies was 0.66 (Fig S2). This suggests that in general, antibodies with a high mRNA–protein correlation in one study are likely to be high in others, whereas antibodies with a low correlation are likely to be low in others.

There are many factors, both technical and biological, that influence the variation in mRNA–protein correlations across proteins (Vogel & Marcotte, 2012; Liu et al, 2016; Buccitelli & Selbach, 2020). However, we reasoned that if less reliable antibodies were the primary cause for the differences observed in Fig 1, then we should not see the same difference between the two groups of proteins when using protein abundance measurements quantified by MS. To assess this, we analysed six CPTAC studies that match the cancer type in TCGA RPPA studies (breast cancer [Mertins et al, 2016], ovarian cancer [Zhang et al, 2016], colorectal adenocarcinoma [Zhang et al, 2014], endometrial carcinoma [Dou et al, 2020], clear cell renal cell carcinoma [Clark et al, 2019], and glioblastoma [Wang et al, 2021]). On repeating the above analysis on the mRNA–protein correlations for

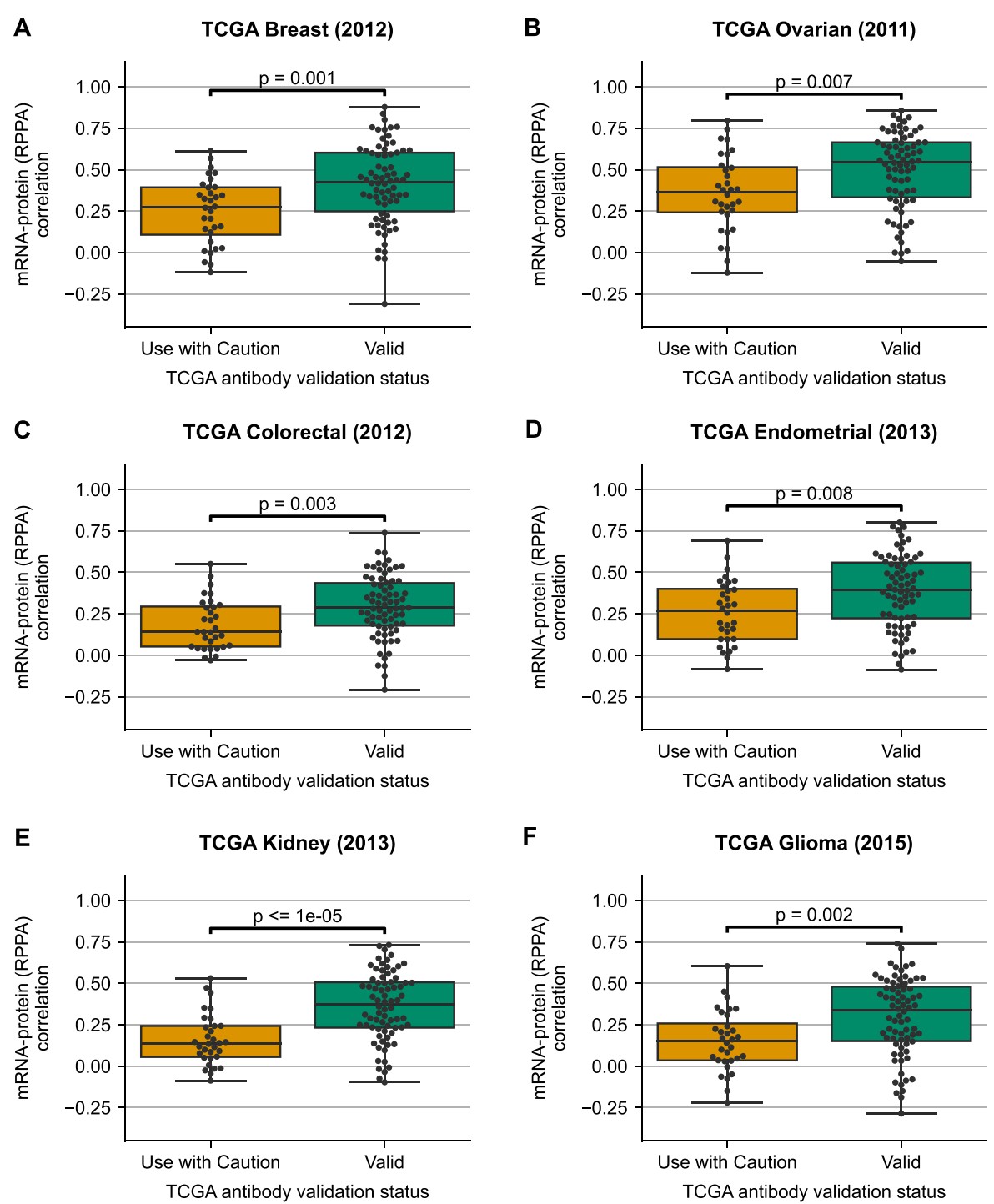

**Figure 1. Proteins measured with less reliable antibodies have significantly lower mRNA–protein correlation.**
Boxplots showing the distribution of mRNA–protein correlations for proteins quantified using antibodies labelled as "Valid" or "Use with Caution."
**(A, B, C, D, E, F)** Correlation is measured using Spearman's correlation across all patients with measurements for both mRNA and protein in breast cancer (Cancer Genome Atlas Network, 2012b) (A), ovarian cancer (Cancer Genome Atlas Research Network, 2011) (B), colorectal adenocarcinoma (Cancer Genome Atlas Network, 2012a) (C), endometrial carcinoma (Cancer Genome Atlas Research Network et al, 2013a) (D), kidney renal cell carcinoma (Cancer Genome Atlas Research Network, 2013) (E), and low-grade glioma (Cancer Genome Atlas Research Network et al, 2015) (F). For each box plot, the black central line represents the median, the top and bottom lines represent the first and third quartiles, and the whiskers extend to 1.5 times the interquartile range past the box. Each point on a box plot represents a protein. Source data are available for this figure.

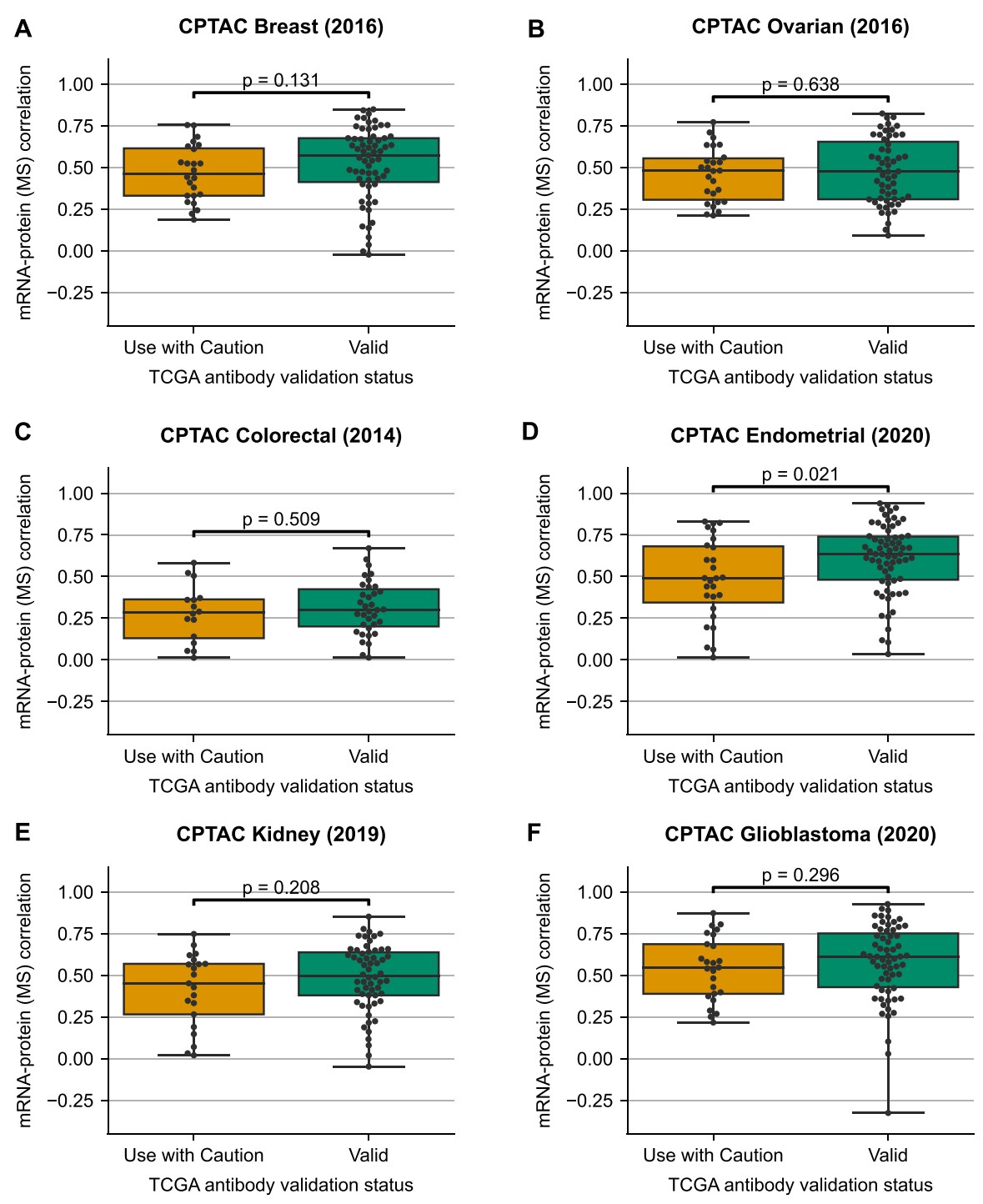

**Figure 2. mRNA–protein correlations for proteins measured by mass spectrometry are not significantly influenced by the associated antibody validation status.**
Boxplots showing the distribution of mRNA–protein correlations wherein proteins are classified based on their antibody validation status and quantified using mass spectrometry. **(A, B, C, D, E, F)** Correlation is measured using Spearman's correlation across all patients with measurements for both mRNA and protein in breast cancer (Mertins et al, 2016) (A), ovarian cancer (Zhang et al, 2016) (B), colorectal adenocarcinoma (Zhang et al, 2014) (C), endometrial carcinoma (Dou et al, 2020) (D), clear cell renal cell carcinoma (Clark et al, 2019) (E), and glioblastoma (Wang et al, 2021) (F). For each box plot, the black central line represents the median, the top and bottom lines represent the first and third quartiles, and the whiskers extend to 1.5 times the interquartile range past the box. Each point on a box plot represents a protein. Source data are available for this figure.

the CPTAC studies, we observed that the two classes of proteins had no particular pattern in the distribution of mRNA–protein correlation (Fig 2). In five out of six studies analysed, there was no significant difference between the two groups, whereas in a single cancer type, there was (P-value = 0.02, Mann–Whitney U test, two-sided).

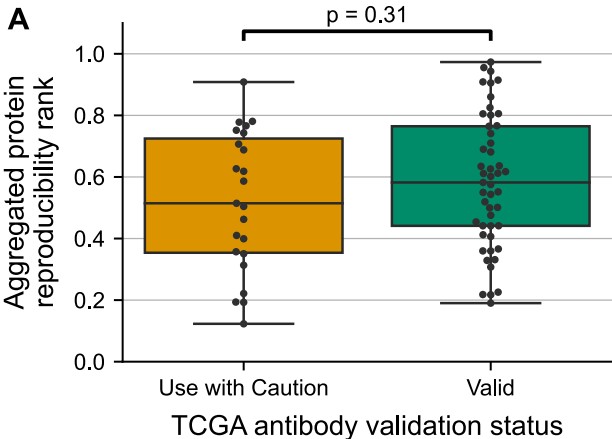

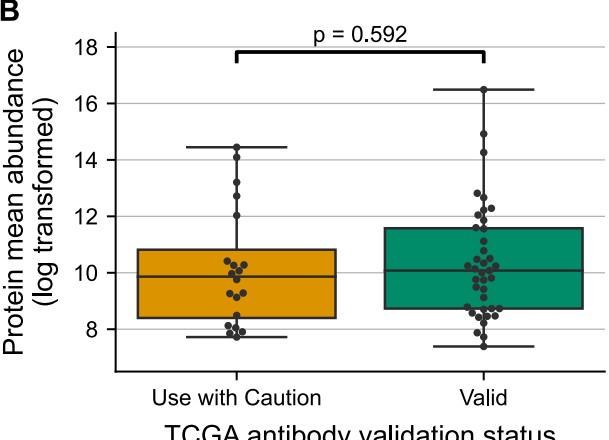

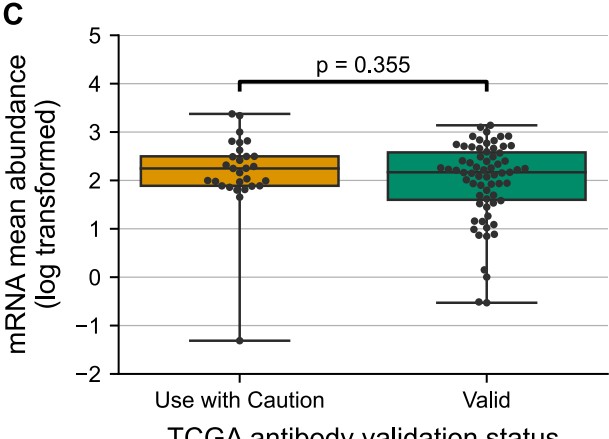

**Figure 3. Antibody reliability does not reflect protein measurement reproducibility, protein abundance or mRNA abundance.**
**(A, B, C)** Boxplots showing the distribution of aggregated protein reproducibility ranks (A), protein abundances measured using MS (B), and mRNA abundances (C) for proteins from TCGA studies classified based on their antibody validation status. For each box plot, the black central line represents the median, the top and bottom lines represent the first and third quartiles, and the whiskers extend to 1.5 times the interquartile range past the box. Each point on a box plot represents a protein.
Source data are available for this figure.

The results in Fig 1 suggest that protein measurements made with antibodies marked as "Use with Caution" have a lower correlation with their cognate mRNA measurements than protein measurements made with "Valid" antibodies. The results in Fig 2 suggest that it is not simply that these proteins display a lower correlation with their cognate mRNAs as the same trend is not evident for MS measurements.

To understand the variance in mRNA–protein correlation that can be associated with antibody validation status, we performed univariate linear regression. The antibody validation status could explain 5.5–18% of the variation in the mRNA–protein correlation for TCGA studies, wherein the protein expressions are measured using RPPA (Fig S3). The average variance explained in the mRNA–protein correlation for TCGA studies is ~9%. However, the average variance explained in the mRNA–protein correlations for the CPTAC studies where proteins are measured using MS is <1% (Fig S3). This suggests that the antibody validation status has an influence on mRNA–protein correlation when protein expression is measured using RPPA.

### Antibody reliability does not reflect protein measurement reproducibility, protein abundance or mRNA abundance

We have previously found that some proteins appear, across multiple studies, to be more reproducibly quantified by MS than others (Upadhya & Ryan, 2022). We exploited this observation to develop an aggregated protein reproducibility rank for each protein by integrating results from three studies with replicate proteomic profiles (Zhang et al, 2016; Vasaikar et al, 2019; Nusinow et al, 2020). The aggregated protein reproducibility rank ranges from 0 to 1 (0—low reproducibility; 1—high reproducibility). Using this score, we found that proteins with more reproducible measurements tended to have higher mRNA–protein correlation across multiple MS studies (Upadhya & Ryan, 2022).

To understand if proteins with less reliable antibodies are also less reproducibly measured by MS, we compared the aggregated protein reproducibility ranks for the proteins with "Valid" and "Use with Caution" antibodies. We observed that there was no significant difference in the distribution of aggregated protein reproducibility ranks for the two groups of proteins (Fig 3A). This suggests that proteins with less reliable antibodies are not more irreproducible when measured using MS and that they are not inherently more challenging to quantify.

We also previously identified that protein abundance influences the reproducibility of protein measurements using MS—more abundant proteins tend to be more reliably quantified by MS (Upadhya & Ryan, 2022). To test if antibodies marked as "Use with Caution" preferentially target proteins with lower abundance, we compared the protein abundances of the targets of these antibodies with those targeted by "Valid" antibodies. We obtained protein abundance measurements from the GTEx project (Jiang et al, 2020), wherein protein abundance is measured using MS-based proteomics. We found that there was no significant difference in protein abundance between the proteins targeted by the two types of antibodies ("Valid" and "Use with Caution") (Fig 3B). The same trend was observed when we assessed mRNA abundances

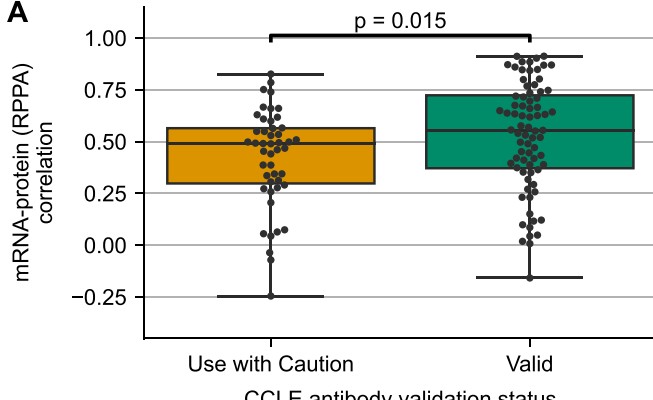

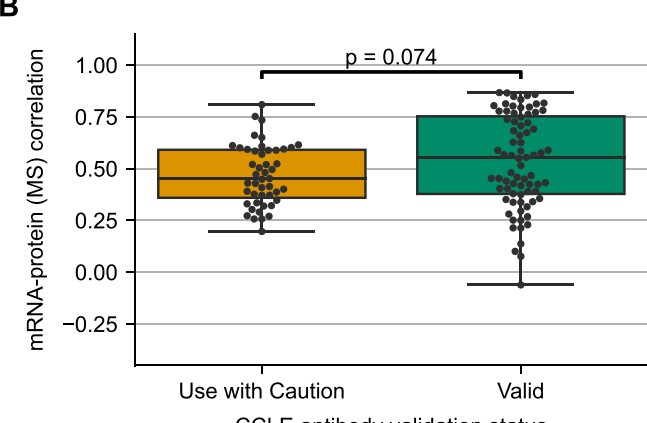

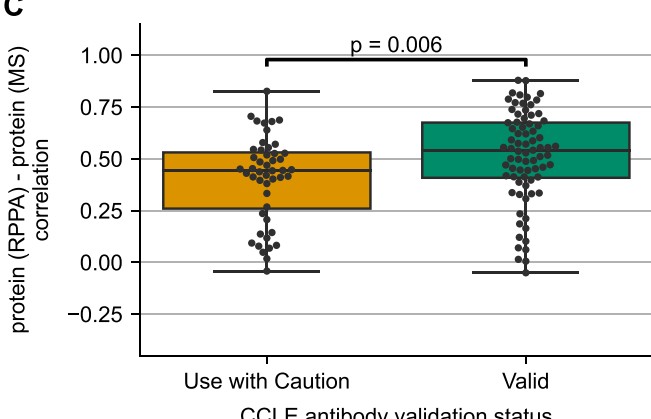

**Figure 4. Validating the influence of antibody validation status on mRNA–protein correlation on CCLE data.**
**(A)** Boxplots showing the distribution of mRNA–protein correlation for proteins quantified using antibodies labelled as "Valid" or "Use with Caution." **(B)** Boxplots showing the distribution of mRNA–protein correlations wherein proteins are classified based on their antibody validation status and quantified using mass spectrometry. **(C)** Boxplots showing the distribution of protein (RPPA)—protein (mass spectrometry) correlations based on their antibody validation status. For each box plot, the black central line represents the median, the top and bottom lines represent the first and third quartiles, and the whiskers extend to 1.5 times the interquartile range past the box. Each point on a box plot represents a protein.
Source data are available for this figure.

obtained from RNA-seq profiles from the GTEx project (Jiang et al, 2020) (Fig 3C).

Overall, these results suggest that the antibodies marked as "Use with Caution" do not target proteins that are less abundant or that are less reproducibly quantified by MS.

### Validation of findings using the cancer cell line encyclopedia

Thus far, we have focussed our analysis on tumour samples profiled by either TCGA or CPTAC. We have used the same, or similar, cancer types when analysing mRNA–protein correlations in both studies. However, a limitation of our analysis is that we have not been able to use the same sets of samples for all analyses—this is because many of the CPTAC MS studies do not have associated RPPA data available, whereas most of TCGA samples profiled by RPPA do not have corresponding MS profiles. Consequently, we have assessed mRNA–protein correlations for RPPA and MS protein quantification using different sets of samples.

The cancer cell line encyclopedia, and associated molecular characterisation efforts, presents an opportunity to validate our findings in an orthogonal dataset where the same cell lines have been profiled using RPPA, MS, and RNA-seq. 359 cancer cell lines have measurements for all three data types available through the DepMap portal (Ghandi et al, 2019; Nusinow et al, 2020). The RPPA data used a panel of 152 antibodies to quantify individual, non-phosphorylated proteins where 79 antibodies overlapped with the TCGA antibodies. We identified that ~38% of the CCLE antibodies were marked as "Use with Caution" (Fig S4). As in our analysis of the TCGA data, we found that proteins quantified using antibodies marked as "Use with Caution" have lower mRNA–protein correlation than those measured with "Valid" antibodies (Fig 4A). Furthermore, as in TCGA, when the same proteins were quantified using MS, there was no significant difference in the mRNA–protein correlation between the two groups (Fig 4B). Similar to TCGA antibodies, the proteins in the CCLE study with "Valid" and "Use with Caution" antibodies had no difference in protein reproducibility, protein or mRNA abundances (Fig S5).

The availability of protein abundance measurements from two different techniques (RPPA and MS) over the same cancer cell lines enables us to directly compare the protein abundance measurements obtained from the two (RPPA and MS) techniques. We computed the correlation between protein abundance measurements obtained from RPPA and MS for each protein and grouped the proteins based on their antibody validation status. We note that there is variability in the observed protein (RPPA)—protein (MS) correlation even for valid antibodies (Fig 4C), with some proteins displaying low correlation across the different measurement techniques, but this is consistent with our previous finding that some proteins may be more reproducibly quantified than others (Upadhya & Ryan, 2022). Despite this variability, we find that proteins with valid antibodies had a significantly higher median protein (RPPA)—protein (MS) correlation compared with the proteins with antibodies labelled "Use with Caution" (Fig 4C) (P-value = $5.920 \times 10^{-3}$, Mann–Whitney U test, two-sided). This suggests that proteins measured with "Use with Caution" antibodies show

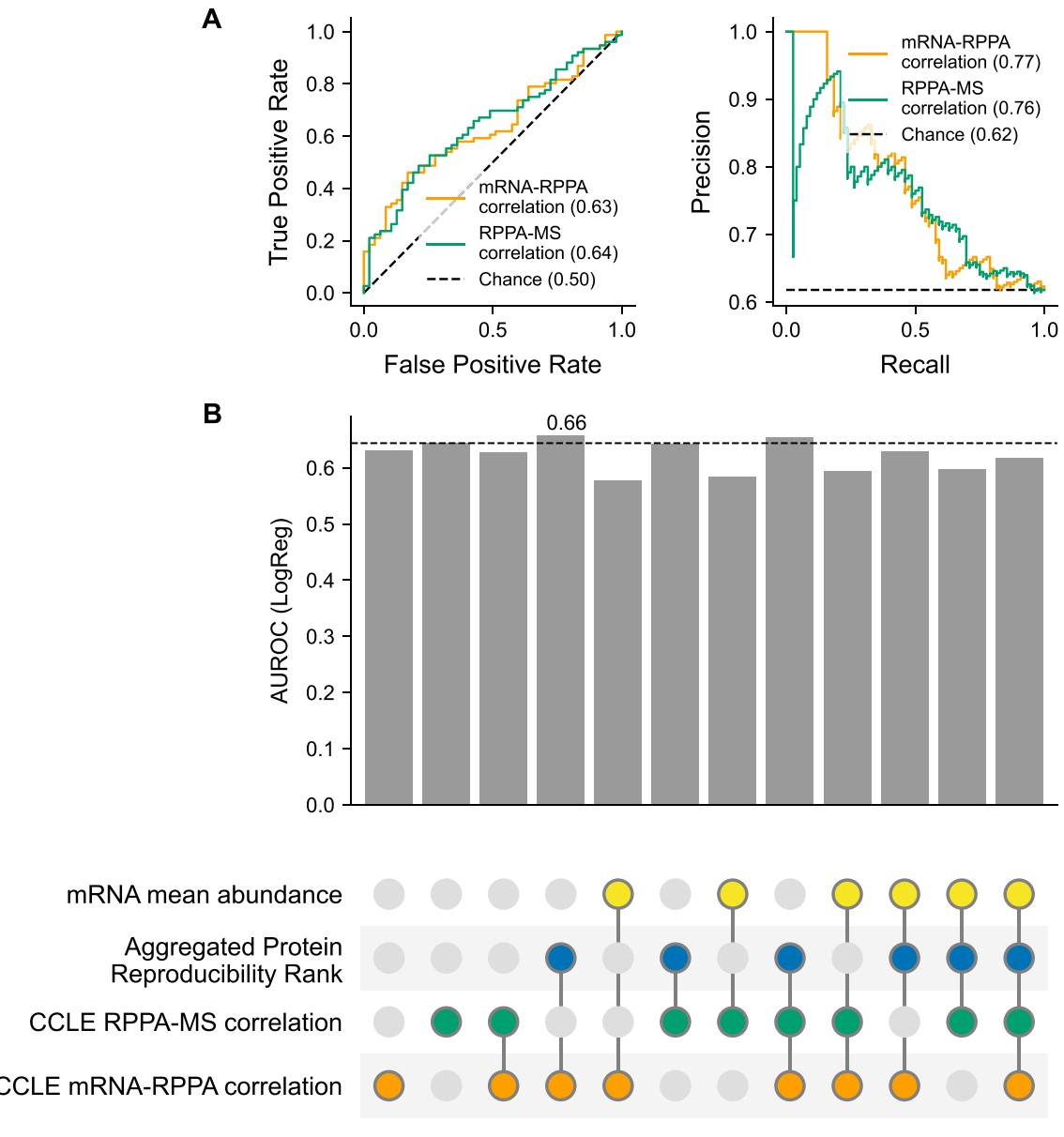

**Figure 5. Assessing the predictive power of multiple features to identify less reliable antibodies of the CCLE dataset.**
**(A)** ROC and precision–recall curves for the logistic regression model predicting the validation status of antibodies using mRNA–RPPA correlation and RPPA–MS correlation over the same set of 123 antibodies. **(B)** For each combination of features indicated using the connected coloured circles, the AUROC score averaged over threefold cross-validation for the logistic regression model is shown in the bar graph. The black dashed line indicates the baseline AUROC score achieved for the RPPA–MS correlation. The highest AUROC score achieved is noted to be 0.66 for the feature set comprising of mRNA–RPPA correlation and aggregated protein reproducibility rank. This analysis was performed over 91 antibodies for which measurements were available for all four features.
Source data are available for this figure.

less concordance with their MS measurements. Overall, this is consistent with these measurements being less reliable and consequently contributing to the lower observed mRNA–protein correlations.

### Assessing the ability to identify less reliable antibodies using mRNA–RPPA and RPPA–MS correlations

The observation that measurements from "Use with Caution" antibodies in general result in lower observed mRNA–RPPA

correlations and lower RPPA–MS correlations suggests that these correlations might be useful for systematically assessing antibody quality. To explore this possibility, we treated distinguishing "Use with Caution" and "Valid" antibodies as a binary classification task and first asked if either individual feature could reliably distinguish the two. Using the CCLE data, we found that mRNA–RPPA correlation had a moderate ability to distinguish between the two classes, with an area under the receiver operating curve (AUROC) of 0.63 and an average precision of 0.77 (Fig 5A). We evaluated the predictive power of RPPA–MS correlations over the same set of proteins and

obtained an AUROC of 0.64 and average precision of 0.76 (Fig 5A). Although these AUROC values are above random expectation (0.5), they suggest that individually these features could not reliably distinguish reliable from less reliable antibodies. This is unsurprising, given that many additional factors influence the reliability of protein measurements—for example, protein abundance, mRNA abundance, and measurement noise (Upadhya & Ryan, 2022). We therefore asked if some of these features might be combined to better distinguish "Use with Caution" from "Valid" antibodies. Given the limited number of training examples, we restricted our analysis to a simple linear classification approach (logistic regression) and evaluated combinations of features together for their classification ability. We assessed combinations of the following four features—mRNA–RPPA correlation, MS–RPPA correlation, mRNA abundance, and aggregated protein reproducibility rank (Fig 5B). We excluded protein abundance as a feature as it contained missing values for many of the antibodies in the evaluation set. Using this approach, we found that it was possible to *slightly* improve on the predictive power of individual features, for example, mRNA–RPPA correlation + aggregated protein reproducibility rank resulted in an AUROC of 0.66. Overall, however, the classification performance was still significantly below what might be used to make confident predictions of antibody reliability.

## Discussion

In previous work, we have demonstrated that some proteins appear to be more reproducibly quantified by MS than others and that more reproducibly quantified proteins tend to have higher observed mRNA–protein correlations (Upadhya & Ryan, 2022). Here, we demonstrate that the mRNA–protein correlations observed using RPPA quantification measurements are significantly influenced by the reliability of the antibodies used. We find that proteins that are quantified by RPPA using more reliable antibodies tend to have higher observed mRNA–protein correlations. This is not true when the same proteins are quantified by MS, suggesting that the cause is likely the antibodies themselves rather than the measured proteins somehow less correlated with their associated mRNAs. Further support for this comes from the observation that RPPA protein measurements are more highly correlated with MS measurements of the same proteins when the RPPA measurements are made using "Valid" antibodies. Although some of the "Valid" antibodies have low RPPA–MS correlation, it may simply reflect proteins that are hard to measure by MS.

Although collectively often referred to as high-quality antibodies, over one-quarter of the antibodies used by TCGA are labelled as "Use with Caution." The list of antibodies, and their validation status, is made available in the supplementary materials of relevant publications (Li et al, 2013; Akbani et al, 2014b) and associated online resources. However, the full set of RPPA measurements, including those made with antibodies marked as "Use with Caution," are still often used for systematic analyses (Zhang et al, 2015, 2017; Koplev et al, 2018; Chen et al, 2019). Furthermore, the measurements made with these antibodies are made available through widely used web resources that facilitate browsing of TCGA

data without any indication that these measurements are less reliable (Cerami et al, 2012; Gao et al, 2013; Vasaikar et al, 2018). Our results suggest that the measurements from these antibodies should be used with caution in systematic analyses and that they should also be flagged as "Use with Caution" in relevant web resources.

We note that the measurements made with antibodies labelled "Use with Caution" may be reproducible but they provide less accurate quantification of the target protein and hence display lower mRNA–protein correlations. It may be the case that they measure the joint abundance of multiple proteins and that this aggregate measurement may be reproducible, but not reliable.

Here, we have focussed on the analysis of RPPA profiles of tumour samples and cancer cell lines, but RPPA profiling is also used in other contexts. For instance, they have been used to systematically profile the responses of cell lines to systematic perturbations (Korkut et al, 2015; Keenan et al, 2018) and also to understand profiles of patients with diseases other than cancer (Napierala et al, 2021). The results of such studies are also likely to be impacted by the use of antibodies of varying qualities.

The issue of antibody reliability is a general challenge for biological studies, not just those that make use of RPPA (Baker, 2015; Goodman, 2018). By performing systematic evaluations of all antibodies used, TCGA RPPA studies make use of antibodies that are likely of significantly higher quality than average. Even those antibodies marked "Use with Caution" are still deemed of sufficient quality for inclusion in assays and are likely to be of higher quality than randomly selected antibodies. Therefore, the trends observed in our analysis are likely to be a lower bound of the potential impact of unreliable antibodies on protein measurements.

The "selection" bias outlined above may in part explain why we have only a moderate ability to use mRNA–RPPA, MS–RPPA, and related measurements to distinguish between reliable and less reliable antibodies (Fig 5B). Another explanation may be that we simply have too little training data to identify the nonlinear patterns or interactions between measurements that might help distinguish between reliable and unreliable antibodies. Our current results suggest that, although "Use with Caution" antibodies result in systematically lower correlations with mRNA and MS measurements, this is not sufficient in itself to assess antibody quality.

## Materials and Methods

### Data collection

The transcriptomic and RPPA data for TCGA PanCancer Atlas studies (breast, ovarian, colorectal, endometrial, kidney, and brain) were downloaded from the USC Xena browser (Goldman et al, 2020). For the analyses using CPTAC studies, the mRNA–protein correlations were obtained from the supplementary table of our previous publication (Upadhya & Ryan, 2022). The transcriptomic profiles in all studies were quantified using RNA-seq. The protein expression profiles of TCGA studies were quantified using RPPA, whereas the CPTAC studies had proteomic

profiles quantified using MS. For the CCLE study, the mRNA expression (Ghandi et al, 2019), MS-based protein expression (Nusinow et al, 2020), and RPPA protein expression profiles (Ghandi et al, 2019) were downloaded from the cancer dependency map portal (https://depmap.org/portal/ccle/).

## Preprocessing protein and transcript expression

The transcript and protein expressions obtained from the USC Xena browser are normalized by TCGA Pan-Cancer project (Thorsson et al, 2018). Thus, no additional data transformation methods were applied. The protein expression profiles quantified using RPPA contained missing values for a small number of proteins. Within each study, we restricted our analyses to proteins that were measured in at least 80% of samples. Some antibodies measure the protein products of multiple genes—for example, the AKT antibody (Akt) measures the total protein abundance from the protein products of the genes *AKT1*, *AKT2*, and *AKT3*. These antibodies that target multiple proteins were excluded from our analysis. Some antibodies are used to measure the abundance of specific phosphoproteins (e.g., 4E-BP1_pS65 measures a specific EIF4EBP1 phosphoprotein) rather than total protein abundances. These were also excluded from our analyses. The criteria for retaining the transcript for the analyses was the same as that for protein—measured (non-zero values) in at least 80% of samples.

## Antibody annotation

The antibodies for TCGA PanCancer Atlas studies were obtained from the supplementary data of Li et al (2013). There were 187 antibodies marked as "Use with Caution" (43), "Validated" (115), and "Under Evaluation" (29). We restricted our analyses to antibodies that are (i) mapped to no more than one protein, (ii) did not map to phosphoproteins (iii) were marked as "Valid" or "Use with Caution" only. Based on the first two criteria, we had 132 antibodies (78 categorised as "Validated," 30 categorised as "Use with Caution," and 24 categorised as "Under Evaluation"). For the antibodies marked as "Under Evaluation," we obtained the current validation status for seven additional antibodies from the validated antibody standard list from MD Anderson Cancer Center website (https://www.mdanderson.org/research/research-resources/core-facilities/functional-proteomics-rppa-core/antibody-information-and-protocols.html). The incorrect or the uncommon gene names in the standard list of antibodies from the afore-mentioned webpage were corrected as indicated on the website. Overall, we were able to analyse 114 proteins with known antibody validation status, that is, "Valid" and "Use with Caution" for TCGA Pan-Cancer Atlas studies (Fig S1 and Table S1). The antibodies that are not used for the analyses in this study have been included in Table S2.

There were 214 antibodies for the CCLE cancer cell line study that were obtained from the supplementary data of Ghandi et al, 2019. Of these 214 antibodies, 139 antibodies were marked as "Valid" and "75" were marked as "Use with Caution" (Fig S4). Based on the criteria for our analyses, the number of antibodies was reduced to 152 antibodies for the cancer cell lines.

## Computation of correlation coefficient

Correlations between mRNA–protein(RPPA) and protein(RPPA)–protein(MS) were computed using the Spearman rank correlation. For each protein in each study, samples with missing values were ignored when computing the correlation. We use Spearman correlation to compute correlations unless explicitly stated.

## Computing protein and mRNA abundances

Protein and mRNA abundances were obtained from the GTEx project (Jiang et al, 2020), wherein protein and mRNA abundances are available for 32 healthy tissues across thousands of proteins and transcripts. Proteins were measured using MS-based proteomics and transcripts were measured using RNA sequencing. Only those proteins/transcripts with abundance measurements present at least across 80% of tissue samples were considered to compute mean abundance. Because the abundance measurements for both proteins and transcripts had a wide range, we further applied $\log_2$ transformation to use the abundance measurements in our analyses.

## Assessing the relationship between antibody validation status and mRNA–protein correlation (Fig S3)

To understand the variance in mRNA–protein correlation explained by antibody validation status, we used linear models given by the equation:

$$c(p) = \alpha + \beta * avs(p),$$

where $c(p)$ is the mRNA–protein (protein expression measured through either RPPA or MS) correlation for each protein, $avs(p)$ is the antibody validation status for each protein represented as 1 if the antibody is "Valid," and 0 if the antibody is marked as "Use with Caution." The coefficients $\alpha$ and $\beta$ are estimated using the ordinary least squares regression method. All linear regressions were carried out using the statsmodel package in Python.

## Assessing the predictive power of multiple features to identify less reliable antibodies

To understand our ability to classify antibodies based on their validation status, we used a logistic regression model for a combination of features—mRNA–RPPA correlation, RPPA–MS correlation, aggregated protein reproducibility rank, and mRNA abundances (Fig 5B). Logistic regression model of the scikit-learn package was trained over the same set of 91 proteins for the different feature combinations. The models were evaluated using the threefold cross-validation approach. The area under the ROC curve and average precision scores were also computed using the scikit-learn package in Python.

## Statistical analysis

All statistical analyses were carried out using Python 3.11.0, Pandas 1.5.2 (McKinney, 2010), numpy 1.24.0 (Harris et al, 2020), scipy 1.9.3 (Virtanen et al, 2020), scikit-learn 1.2.1 (Pedregosa et al, 2011), and statsmodels 0.13.5 (Seabold & Perktold, 2010). The figures were created with Matplotlib 3.6.2 (Hunter, 2007) and Seaborn 0.12.1 (Waskom, 2021).

# Data Availability

This article analysed existing, publicly available data. All original code has been deposited at GitHub (https://github.com/cancergenetics/antibody_quality_limitations.git).

# Supplementary Information

# Acknowledgements

SR Upadhya was funded through the School of Computer Science, University College Dublin and CJ Ryan was funded by the Irish Research Council Laureate Awards 2017/2018. We thank members of the Ryan lab for careful reading of the manuscript and helpful feedback.

## Author Contributions

SR Upadhya: conceptualization, data curation, software, formal analysis, visualization, methodology, and writing—original draft.
CJ Ryan: conceptualization, supervision, funding acquisition, methodology, project administration, and writing—original draft, review, and editing.

## Conflict of Interest Statement

The authors declare that they have no conflict of interest.

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
