## [Reviewer comments · Life Science Alliance]

Life Science Alliance

Antibody reliability influences observed mRNA-protein correlations in tumour samples

Swathi Ramachandra Upadhy and Colm Ryan

DOI: <https://doi.org/10.26508/lsa.202201885>

Corresponding author(s): Colm Ryan, University College Dublin

Review Timeline:

Submission Date:	2022-12-22
Editorial Decision:	2023-02-02
Revision Received:	2023-04-21
Editorial Decision:	2023-04-24
Revision Received:	2023-05-02
Accepted:	2023-05-02

Transaction Report:

February 2, 2023

Re: Life Science Alliance manuscript #LSA-2022-01885

Dr. Colm J. Ryan
University College Dublin
School of Computer Science, Conway Institute of Biomolecular and Biomedical Research and Systems Biology Ireland
University College Dublin, Belfield, Dublin
Dublin, Ireland

Dear Dr. Ryan,

Thank you for submitting your manuscript entitled "Antibody reliability influences observed mRNA-protein correlations in tumour samples" to Life Science Alliance. The manuscript was assessed by expert reviewers, whose comments are appended to this letter. We invite you to submit a revised manuscript addressing the Reviewer comments.

Thank you for this interesting contribution to Life Science Alliance. We are looking forward to receiving your revised manuscript.

Sincerely,

B. MANUSCRIPT ORGANIZATION AND FORMATTING:

Reviewer #1 (Comments to the Authors (Required)):

Transcriptomic quantitation methods are cost-effective and highly reliable, and thus transcriptional levels are often used as a surrogate for protein levels. However, the degree to which mRNA levels correlate with protein levels has been a contentious topic, and subject to significant debate. Upadhy & Ryan previously demonstrated that measurement reproducibility accounts for a substantial fraction of the variation between mRNA and protein abundances. Here, this same team extends these findings and demonstrate that antibody quality also affects mRNA-protein correlation for Reverse Phase Protein Arrays (RPPA)- an antibody-dependent proteomic method that has been extensively used to quantify hundreds of proteins in thousands of tumors by the Cancer Genome Atlas (TCGA). In brief, they stratify antibodies by the TCGA quality metric "Use with caution" versus "valid" (based on western blot quality and RPPA-western correlation). They then examine mRNA-protein correlation using proteomic data from either RPPA experiments (generated with these two groups of antibodies) or Mass Spectrometry (MS) data (which is antibody-independent). They show that the antibody quality improves mRNA-Protein correlation for RPPA but not MS proteomics. Overall, the manuscript is well written, and the study will be useful to the research community, and this reviewer has only minor comments that need to be addressed and suggestions that can potentially increase impact.

-Can the authors dig a bit more into the data to try and salvage some "use with caution antibodies"? That is, some of the RPPA-mRNA correlations using "caution" antibodies are quite high. Are these the same proteins that correlate well in RPPA-MS comparisons? Could this (RPPA-MS) be a way to refine antibody QC? With "valid" antibodies against proteins with poor RPPA-MS correlations - is this a reason to be concerned?

-The title of Figure 1 and section 1 implies that the authors performed a quality control of the TCGA antibodies "About one third of the antibodies used to quantify proteins in the TCGA RPPA Pan-Cancer dataset are not reliable". But the authors are simply using the TCGA quality control criteria, and this is not a result of the current study. The filtering criteria should be described (removing phospho proteins etc), but this should not be written up as a section in results and titled as such unless the authors contribute new information about these antibodies.

-The individual studies/datasets that are used in figures 2, 3 (Breast 2012) should be cited in figure legends.

-Line 190: it would be helpful to briefly introduce the method used to generate the reproducibility ranking (based on replicates etc) cited from their 2022 paper.

-Line 217: GTEX data- is this MS based (should be mentioned in legend also).

-Line 221: Figure 4C - what is source of this data?

Reviewer #2 (Comments to the Authors (Required)):

In their paper, „Antibody reliability influences observed mRNA-protein correlations in tumour samples", Upadhy and Ryan investigate if the antibody classification as "use with caution" in large-scale RPPA data sets of tumors is correlated with impaired quantification of proteins. They do so by mainly using mRNA-protein correlation as proxy for quantifiability.

Overall, this is of course an important problem to assess the data quality of those widely used data sets. And indeed they convincingly show that those antibodies labeled as "use with caution" perform, on average, worse as they show significantly lower correlations between RNA and protein signals. This is not - or at least not to the extend - evident in mass-spec data. They also show that the validation classes are not significantly different with respect to "reproducibility", mean protein and mRNA abundance, and reproduce some of their results using data on cell lines (where both RPPA and mass spec data are available).

One major concern that I have is that all they show is the "on average" behavior, while individual antibodies of each class show strongly diverging behavior. This is most evident in Figure 5D (which shows that some validated antibodies do not correlate at all with the mass-spec derived protein signal). So it is not only the class that influences performance. I would suggest to use that data (in Fig. 5D) to really dissect the causes of quantitative failure. How does correlation between RPPA and proteomics depend

on protein level, mRNA level, antibody validation status and so on. And with this really compile a list of high fidelity antibodies and apply them to the other cohorts.

Minor:

- One wonders if those antibodies that perform well in one study are also performing well in another (e.g. in 3, are those antibodies that have high correlation the same in all the studies?)
- The materials and methods section leaves out essential details: How are the raw data transformed (if they are), are they normalized?

We thank both reviewers for their positive assessment of our manuscript and their helpful feedback. We provide a point-by-point response below. Reviewer's comments are in black, and our responses are in blue.

Reviewers' comments with responses:

Reviewer #1: Transcriptomic quantitation methods are cost-effective and highly reliable, and thus transcriptional levels are often used as a surrogate for protein levels. However, the degree to which mRNA levels correlate with protein levels has been a contentious topic, and subject to significant debate. Upadhyya & Ryan previously demonstrated that measurement reproducibility accounts for a substantial fraction of the variation between mRNA and protein abundances. Here, this same team extends these findings and demonstrate that antibody quality also affects mRNA-protein correlation for Reverse Phase Protein Arrays (RPPA)- an antibody-dependent proteomic method that has been extensively used to quantify hundreds of proteins in thousands of tumors by the Cancer Genome Atlas (TCGA). In brief, they stratify antibodies by the TCGA quality metric "Use with caution" versus "valid" (based on western blot quality and RPPA-western correlation). They then examine mRNA-protein correlation using proteomic data from either RPPA experiments (generated with these two groups of antibodies) or Mass Spectrometry (MS) data (which is antibody-independent). They show that the antibody quality improves mRNA-Protein correlation for RPPA but not MS proteomics. Overall, the manuscript is well written, and the study will be useful to the research community, and this reviewer has only minor comments that need to be addressed and suggestions that can potentially increase impact.

Our response: We thank the reviewer for the positive feedback on our manuscript. Responses to specific queries are given below.

-Can the authors dig a bit more into the data to try and salvage some "use with caution antibodies"? That is, some of the RPPA-mRNA correlations using "caution" antibodies are quite high. Are these the same proteins that correlate well in RPPA-MS comparisons? Could this (RPPA-MS) be a way to refine antibody QC?

Our response: We thank the reviewer for the suggestion. We have added a new section to the manuscript (Page 10) addressing this question which we have pasted below:

Assessing the ability to identify less reliable antibodies using mRNA-RPPA and RPPA-MS correlations

The observation that measurements from 'Use with Caution' antibodies in general result in lower observed mRNA-RPPA correlations and lower RPPA-MS correlations suggests that these correlations might be useful for systematically assessing antibody quality. To explore this possibility, we treated distinguishing 'Use with Caution' and 'Valid' antibodies as a binary classification task and first asked if either individual feature could reliably distinguish the two. Using the CCLE data we found that mRNA-RPPA correlation had a moderate ability to distinguish between the two classes, with an area under the receiver operating curve (AUROC) of 0.63 and average precision of 0.77 (Fig 5A). We evaluated the predictive power of RPPA-MS correlations over the same set of proteins and obtained an AUROC of 0.64 and average precision of 0.76 (Fig 5A). While these AUROC values are above random expectation (0.5) they suggest that individually these features could not reliably distinguish reliable from less reliable antibodies. This is unsurprising, given that many additional factors influence the reliability of protein measurements – e.g. protein abundance, mRNA abundance, and measurement noise (Upadhyya & Ryan, 2022). We therefore asked if some of these features might be combined to better distinguish 'Use with Caution' from 'Valid' antibodies. Given the limited number of training examples, we restricted our analysis to a simple linear classification approach (logistic regression) and evaluated combinations of features together for their classification ability. We assessed combinations of the following four

features – mRNA-RPPA correlation, MS-RPPA correlation, mRNA abundance, and Aggregated Protein Reproducibility Rank (Fig 5B). We excluded protein abundance as a feature as it contained missing values for many of the antibodies in the evaluation set. Using this approach we found that it was possible to slightly improve on the predictive power of individual features, e.g. mRNA-RPPA correlation + Aggregated Protein Reproducibility Rank resulted in an AUROC of 0.66. Overall, however, the classification performance was still significantly below what might be used to make confident predictions of antibody reliability.

Figure 5. Assessing the predictive power of multiple features to identify less reliable antibodies of the CCLE dataset

(A) ROC and Precision-Recall curves for the logistic regression model predicting the validation status of antibodies using mRNA-RPPA correlation and RPPA-MS correlation over the same set of 123 antibodies. (B) For each combination of features indicated using the connected coloured circles, the AUROC score averaged over 3-fold cross-validation for the logistic regression model is shown in the bar graph. The black dashed line indicates the baseline AUROC score achieved for the RPPA-MS correlation. The highest AUROC score achieved is noted to be 0.66 for the feature set comprising of mRNA-RPPA correlation and Aggregated Protein Reproducibility Rank. This analysis was performed over 91 antibodies for which measurements were available for all four features.

We note that this is also discussed in the discussion (Page 12) as follows with the new text in bold:

The issue of antibody reliability is a general challenge for biological studies, not just those that make use of RPPA (Baker, 2015; Goodman, 2018). By performing systematic evaluations of all antibodies

used, the TCGA RPPA studies make use of antibodies that are likely of significantly higher quality than average. Even those antibodies marked 'Use with Caution' are still deemed of sufficient quality for inclusion in assays and are likely to be of higher quality than randomly selected antibodies. Therefore the trends observed in our analysis are likely to be a lower bound of the potential impact of unreliable antibodies on protein measurements.

The 'selection' bias outlined above may in part explain why we have only a moderate ability to use mRNA-RPPA, MS-RPPA and related measurements to distinguish between reliable and less reliable antibodies (Fig 5B). Another explanation may be that we simply have too little training data to identify the non-linear patterns or interactions between measurements that might help distinguish between reliable and unreliable antibodies. Our current results suggest that, while 'Use with Caution' antibodies result in systematically lower correlations with mRNA and MS measurements, this is not sufficient in itself to assess antibody quality.

With "valid" antibodies against proteins with poor RPPA-MS correlations - is this a reason to be concerned?

We do not think that there is a reason to be concerned with valid antibodies that have low RPPA-MS correlations as we note that we have previously observed significant variation in the reliability of MS measurements for different proteins (Upadhy and Ryan, Cell Reports Methods 2022). Low RPPA-MS correlations may simply reflect proteins that are hard to measure by MS. We have addressed this in the text as follows (Page 10):

We note that there is variability in the observed protein (RPPA) – protein (MS) correlation even for valid antibodies (Fig 4C), with some proteins displaying low correlation across the different measurement techniques, but this is consistent with our previous finding that some proteins may be more reproducibly quantified than others (Upadhy & Ryan, 2022).

-The title of Figure 1 and section 1 implies that the authors performed a quality control of the TCGA antibodies "About one third of the antibodies used to quantify proteins in the TCGA RPPA Pan-Cancer dataset are not reliable". But the authors are simply using the TCGA quality control criteria, and this is not a result of the current study. The filtering criteria should be described (removing phospho proteins etc), but this should not be written up as a section in results and titled as such unless the authors contribute new information about these antibodies.

Our response: We agree with the reviewer, and have addressed this by moving some of the relevant text, below, from the first results section to the introduction (Page 2):

Although collectively they are often referred to as 'high-quality antibodies' (Akbari et al, 2014b; Zhang et al, 2017; Şenbabaoğlu et al, 2016; Cancer Genome Atlas Research Network, 2017), the quality of the antibodies used for TCGA RPPA studies vary. All antibodies used are assessed by the MD Anderson Cancer Center (Li et al, 2013; Akbari et al, 2014b; Chen et al, 2019). The two minimum criteria for validating antibody specificity used by MD Anderson are (i) a single or dominant band in a Western blot around the expected molecular weight of the target protein and (ii) a good Pearson correlation (>0.7) between abundances measured by RPPA and Western blotting across multiple samples (Li et al, 2013; Akbari et al, 2014a). Based on these criteria antibodies are either discarded as unfit for use, categorised as 'Valid', or categorised as 'Use with Caution'. The antibodies marked as 'Valid' indicate that they bind to the intended target protein while the ones marked as 'Use with Caution' indicate they may bind to off-target or multiple proteins along with the target protein. Although the performance of 'Use with Caution' antibodies is poorer than those categorised as 'Valid', they are still used for quantification, typically because they bind to a protein known to have an important role in cancer.

We have moved the relevant figure to the supplement and now reference it as follows in the results (Page 3):

Approximately one-quarter (27%) of the antibodies used for the TCGA RPPA studies are labelled as 'Use with Caution' (Li et al, 2013). The reliability of the antibody used to quantify protein abundances will impact all downstream analyses of protein measurements including the analysis of mRNA-protein correlations. To understand the impact of antibody reliability on observed mRNA-protein correlations we obtained RPPA measurements from the TCGA Pan-Cancer study (Thorsson et al, 2018). This dataset contains measurements for 258 proteins and phosphoproteins. However, the antibody reliability information is available for only 187 proteins and phosphoproteins (Li et al, 2013). Among these 187 antibodies, we further restricted our analyses to those that are annotated as measuring the abundance of a single, non-phosphorylated protein (See Materials and Methods section). Of the 114 antibodies in this category, 34 are labelled as 'Use with Caution' while 80 are labelled as 'Valid', i.e. ~30% of antibodies should be used with caution (Fig S1, Table S1). The 'Use with Caution' antibodies in this set include antibodies that bind the protein products of frequently altered cancer driver genes, such as the oncogenes MYC and BRAF as well as the tumour suppressors BRCA2 and VHL.

The relevant figures in the supplement are now titled '**Figure S1. Reliability of antibodies from the TCGA RPPA Pan-Cancer dataset analysed in this study**' and '**Figure S4. Reliability of antibodies from the CCLE dataset analysed in this study**'

-The individual studies/datasets that are used in figures 2, 3 (Breast 2012) should be cited in figure legends.

Our response: We agree with the reviewer, and have now cited the relevant studies in the figure legends.

Additionally, we have edited the title of each subplot to contain 'TCGA' / 'CPTAC' keywords to clarify the studies being used in the figures. An example of the modified subplot in the current Figures 1 and 2 are shown above.

-Line 190: it would be helpful to briefly introduce the method used to generate the reproducibility ranking (based on replicates etc) cited from their 2022 paper.

Our response: We agree with the reviewer, and have addressed the text change (Page 7) as follows.

We have previously found that some proteins appear, across multiple studies, to be more reproducibly quantified by mass spectrometry than others (Upadhy & Ryan, 2022). We exploited this observation to develop an aggregated protein reproducibility rank for each protein by integrating results from three studies with replicate proteomic profiles (Nusinow et al, 2020; Vasaiakar et al, 2019; Zhang et al, 2016). The aggregated protein reproducibility rank ranges from 0 to 1 (0 – low reproducibility; 1 – high reproducibility). Using this score we found that proteins with more reproducible measurements tended to have higher mRNA-protein correlation across multiple mass spectrometry studies (Upadhy & Ryan, 2022).

-Line 217: GTEx data- is this MS based (should be mentioned in legend also).

Our response: We have addressed this comment in the current Figure 3 legend and also in the text (Page 7, line 203) as follows:

We obtained protein abundance measurements from the GTEx project (Jiang et al, 2020) wherein protein abundance is measured using mass spectrometry-based proteomics.

-Line 221: Figure 4C - what is source of this data?

Our response: We obtained the mRNA abundances also from GTEx study. We have clarified this in the text (Page 7, line 208) as follows:

The same trend was observed when we assessed mRNA abundances obtained from RNASeq profiles from the GTEx project (Jiang et al, 2020) (Fig 3C).

In addition to this, we have also included a detailed section in the Materials and Methods section, *Computing protein and mRNA abundances* (Page 15) as follows.

Computing protein and mRNA abundances

Protein and mRNA abundances were obtained from the GTEx project (Jiang et al, 2020) wherein protein and mRNA abundances are available for 32 healthy tissues across thousands of proteins and transcripts. Proteins were measured using mass spectrometry-based proteomics and transcripts were measured using RNA sequencing. Only those proteins/transcripts with abundance measurements present at least across 80% of tissue samples were considered to compute mean abundance. Since the abundance measurements for both proteins and transcripts had a wide range, we further applied log2 transformation to use the abundance measurements in our analyses.

End of Reviewer 1 comments

Reviewer #2:

In their paper, "Antibody reliability influences observed mRNA-protein correlations in tumour samples", Upadhy and Ryan investigate if the antibody classification as "use with caution" in large-scale RPPA data sets of tumors is correlated with impaired quantification of proteins. They do so by mainly using mRNA-protein correlation as proxy for quantifiability.

Overall, this is of course an important problem to assess the data quality of those widely used data sets. And indeed they convincingly show that those antibodies labeled as "use with caution" perform, on average, worse as they show significantly lower correlations between RNA and protein signals. This is not - or at least not to the extent - evident in mass-spec data. They also show that the validation classes are not significantly different with respect to "reproducibility", mean protein and mRNA abundance, and reproduce some of their results using data on cell lines (where both RPPA and mass spec data are available).

Our response: We thank the reviewer for the positive feedback on our manuscript. The major and minor concerns are addressed below.

One major concern that I have is that all they show is the "on average" behavior, while individual antibodies of each class show strongly diverging behavior. This is most evident in Figure 5D (which shows that some validated antibodies do not correlate at all with the mass-spec derived protein signal). So it is not only the class that influences performance. I would suggest to use that data (in Fig. 5D) to really dissect the causes of quantitative failure. How does correlation between RPPA and proteomics depend on protein level, mRNA level, antibody validation status and so on. And with this really compile a list of high fidelity antibodies and apply them to the other cohorts.

Our response: We thank the reviewer for the suggestion. We have added a new section to the manuscript (Page 10) addressing this question which we have pasted below:

Assessing the ability to identify less reliable antibodies using mRNA-RPPA and RPPA-MS correlations

The observation that measurements from 'Use with Caution' antibodies in general result in lower observed mRNA-RPPA correlations and lower RPPA-MS correlations suggests that these correlations might be useful for systematically assessing antibody quality. To explore this possibility, we treated distinguishing 'Use with Caution' and 'Valid' antibodies as a binary classification task and first asked if either individual feature could reliably distinguish the two. Using the CCLE data we found that mRNA-RPPA correlation had a moderate ability to distinguish between the two classes, with an area under the receiver operating curve (AUROC) of 0.63 and average precision of 0.77 (Fig 5A). We evaluated the predictive power of RPPA-MS correlations over the same set of proteins and obtained an AUROC of 0.64 and average precision of 0.76 (Fig 5A). While these AUROC values are above random expectation (0.5) they suggest that individually these features could not reliably distinguish reliable from less reliable antibodies. This is unsurprising, given that many additional factors influence the reliability of protein measurements – e.g. protein abundance, mRNA abundance, and measurement noise (Upadhy & Ryan, 2022). We therefore asked if some of these features might be combined to better distinguish 'Use with Caution' from 'Valid' antibodies. Given the limited number of training examples, we restricted our analysis to a simple linear classification approach (logistic regression) and evaluated combinations of features together for their classification ability. We assessed combinations of the following four features – mRNA-RPPA correlation, MS-RPPA correlation, mRNA abundance, and Aggregated Protein Reproducibility Rank (Fig 5B). We excluded protein abundance as a feature as it contained missing values for many of the antibodies in the evaluation set. Using this approach we found that it was possible to slightly improve on the predictive power of individual features, e.g. mRNA-RPPA correlation + Aggregated Protein Reproducibility Rank resulted in an AUROC of 0.66. Overall, however, the

classification performance was still significantly below what might be used to make confident predictions of antibody reliability.

Figure 5. Assessing the predictive power of multiple features to identify less reliable antibodies of the CCLE dataset

(A) ROC and Precision-Recall curves for the logistic regression model predicting the validation status of antibodies using mRNA-RPPA correlation and RPPA-MS correlation over the same set of 123 antibodies. (B) For each combination of features indicated using the connected coloured circles, the AUROC score averaged over 3-fold cross-validation for the logistic regression model is shown in the bar graph. The black dashed line indicates the baseline AUROC score achieved for the RPPA-MS correlation. The highest AUROC score achieved is noted to be 0.66 for the feature set comprising of mRNA-RPPA correlation and Aggregated Protein Reproducibility Rank. This analysis was performed over 91 antibodies for which measurements were available for all four features.

We note that this is also discussed in the discussion (Page 12) as follows with the new text in bold:

The issue of antibody reliability is a general challenge for biological studies, not just those that make use of RPPA (Baker, 2015; Goodman, 2018). By performing systematic evaluations of all antibodies used, the TCGA RPPA studies make use of antibodies that are likely of significantly higher quality than average. Even those antibodies marked 'Use with Caution' are still deemed of sufficient quality for inclusion in assays and are likely to be of higher quality than randomly selected antibodies. Therefore the trends observed in our analysis are likely to be a lower bound of the potential impact of unreliable antibodies on protein measurements.

The 'selection' bias outlined above may in part explain why we have only a moderate ability to use mRNA-RPPA, MS-RPPA and related measurements to distinguish between reliable and less reliable antibodies (Fig 5B). Another explanation may be that we simply have too little training data to identify the non-linear patterns or interactions between measurements that might help distinguish between reliable and unreliable antibodies. Our current results suggest that, while 'Use with Caution' antibodies result in systematically lower correlations with mRNA and MS measurements, this is not sufficient in itself to assess antibody quality.

Minor:

- One wonders if those antibodies that perform well in one study are also performing well in another (e.g. in 3, are those antibodies that have high correlation the same in all the studies?)

Our response: We now address this in the text as follows (Page 3):

To assess if mRNA-protein correlations for different antibodies were consistent across studies, we computed the Pearson's correlation between the mRNA-protein correlations measured in each pair of TCGA studies. We found that the average correlation between all pairs of studies was 0.66 (Fig S2). This suggests that in general antibodies with a high mRNA-protein correlation in one study are likely to be high in others, while antibodies with a low correlation are likely to be low in others.

The relevant figure is included below:

Fig S2. Assessing the consistency in antibody derived mRNA-protein correlations across TCGA PanCancer studies.

Heatmap showing the Pearson correlation between the mRNA-protein correlations derived in pairs of TCGA studies. Gene-wise mRNA-protein correlations were calculated using RPPA measurements for each study, and the resulting values were then compared across pairs of studies using Pearson's correlation.

- The materials and methods section leaves out essential details: How are the raw data transformed (if they are), are they normalized?

Our response: We did not normalize the transcript or protein expression used ourselves, rather we made use of the normalised versions provided by the TCGA Pan-Cancer project. This is now clarified in the text (Page 14) as follows:

Pre-processing protein and transcript expression

The transcript and protein expressions obtained from the USC Xena browser are normalized by the TCGA Pan-Cancer project (Thorsson et al, 2018). Thus, no additional data transformation methods were applied. The protein expression profiles quantified using RPPA contained missing values for a small number of proteins. Within each study, we restricted our analyses to proteins that were measured in at least 80% of samples. Some antibodies measure the protein products of multiple genes – e.g. the AKT antibody (Akt) measures the total protein abundance from the protein products of the genes AKT1, AKT2, and AKT3. These antibodies that target multiple proteins were excluded from our analysis. Some antibodies are used to measure the abundance of specific phosphoproteins (e.g. 4E-BP1_pS65 measures a specific EIF4EBP1 phosphoprotein) rather than total protein abundances. These were also excluded from our analyses. The criteria for retaining the transcript for the analyses was the same as that for protein – measured (non-zero values) in at least 80% of samples.

However, we have processed the mRNA and protein abundances used to do the analyses in Figure 3 and Figure S5. Therefore, we have added a new section in the Materials and Methods section, *Computing protein and mRNA abundances* (Page 15) as follows:

Computing protein and mRNA abundances

Protein and mRNA abundances were obtained from the GTEx project (Jiang et al, 2020) wherein protein and mRNA abundances are available for 32 healthy tissues across thousands of proteins and transcripts. Proteins were measured using mass spectrometry-based proteomics and transcripts were measured using RNA sequencing. Only those proteins/transcripts with abundance measurements present at least across 80% of tissue samples were considered to compute mean abundance. Since the abundance measurements for both proteins and transcripts had a wide range, we further applied log2 transformation to use the abundance measurements in our analyses.

End of Reviewer 2 comments

April 24, 2023

RE: Life Science Alliance Manuscript #LSA-2022-01885R

Dr. Colm J. Ryan
University College Dublin
School of Computer Science, Conway Institute of Biomolecular and Biomedical Research and Systems Biology Ireland
University College Dublin, Belfield, Dublin 4
Dublin, Dublin D04V1W8
Ireland

Dear Dr. Ryan,

Thank you for submitting your revised manuscript entitled "Antibody reliability influences observed mRNA-protein correlations in tumour samples". We would be happy to publish your paper in Life Science Alliance pending final revisions necessary to meet our formatting guidelines.

- please upload your manuscript text file as a doc file
- please upload both your main and supplementary figures as single files and add a separate figure legend section to your main manuscript text

A. FINAL FILES:

B. MANUSCRIPT ORGANIZATION AND FORMATTING:

Thank you for your attention to these final processing requirements. Please revise and format the manuscript and upload materials within 3 days.

Sincerely,

Reviewer #1 (Comments to the Authors (Required)):

The authors have addressed all of my concerns.

May 2, 2023

RE: Life Science Alliance Manuscript #LSA-2022-01885RR

Dr. Colm J. Ryan
University College Dublin
School of Computer Science, Conway Institute of Biomolecular and Biomedical Research and Systems Biology Ireland
University College Dublin, Belfield, Dublin 4
Dublin, Dublin D04V1W8
Ireland

Dear Dr. Ryan,

Thank you for submitting your Research Article entitled "Antibody reliability influences observed mRNA-protein correlations in tumour samples". It is a pleasure to let you know that your manuscript is now accepted for publication in Life Science Alliance. Congratulations on this interesting work.

DISTRIBUTION OF MATERIALS:

Again, congratulations on a very nice paper. I hope you found the review process to be constructive and are pleased with how the manuscript was handled editorially. We look forward to future exciting submissions from your lab.

Sincerely,
